# Impact of a Nutrition Knowledge Intervention on Knowledge and Food Behaviour of Women Within a Rural Community

**DOI:** 10.3390/nu16234107

**Published:** 2024-11-28

**Authors:** Queen E. M. Mangwane, Abdulkadir Egal, Delia Oosthuizen

**Affiliations:** 1Department of Hospitality Management, Faculty of Management Sciences, Tshwane University of Technology, Pretoria 0001, South Africa; 2Department of Tourism & Integrated Communication, Faculty of Human Sciences, Vaal University of Technology, Vanderbijlpark 1900, South Africa; egal@ieaso.so (A.E.); deliao@vut.ac.za (D.O.)

**Keywords:** nutrition education, knowledge, caregivers, nutrition, behaviour, women

## Abstract

Introduction: The influence of women in shaping household dietary habits is undeniable, with their maternal nutritional knowledge significantly impacting the overall well-being of their families. The study objective was to evaluate the impact of maternal nutritional knowledge on household dietary habits, emphasising the improvement in women’s nutrition knowledge and food-related behaviours. Purpose: This study aimed to assess the immediate and long-term effects of a nutrition education intervention on the nutrition knowledge and food-related behaviours of women. Methodology: A quasi-experimental research design was employed to assess the effects of a Nutrition Education Programme (NEP) on female caregivers’ nutrition knowledge and food-related behaviours. This study followed a four-phase approach: baseline survey, programme formulation, implementation and evaluation. Data were collected at various stages of this study using two measurement tools: a Dietary Diversity Questionnaire and a Nutrition Knowledge Questionnaire. Results: Statistical analysis was performed to answer the research questions and to test the null hypotheses at a significance level of 0.05. The intervention positively affected nutrition knowledge, with the mean percentage increasing from 49.1% to 63.7% and reaching 64.4% at follow-up, indicating long-term knowledge retention. The findings of this study suggest a positive shift towards increased consumption of nutrient-rich food groups. There were significant improvements in the food groups’ consumption of the meat group (7.15 ± 2.35), eggs (1 ± 0), dairy (3.76 ± 1.19), cereal (8.78 ± 2.09), legumes (2.86 ± 0.95) and fats and oils (2.12 ± 0.55). Additionally, the variety of food groups consumed significantly improved (*p* = 0.012) post-intervention, with an increasing trend in the consumption of a variety of food groups (7–9). Recommendation: Tailored nutrition education (NE) programs, in conjunction with addressing socioeconomic barriers positively impact nutritional behaviours, promote healthier food consumption patterns and assist in long-term knowledge retention in disadvantaged communities.

## 1. Introduction

A global effort to combat malnutrition among vulnerable groups, and the underserved communities, has led to the emergence of highly effective interventions [1,2] aimed at addressing the challenges through community engagement and delivery strategies [3,4,5]. Nutrition education (NE) has been widely adopted as a strategy to inform and modify dietary habits while addressing malnutrition and related risk factors in populations [6,7,8]. The goal of NE is to empower individuals to recognise their nutritional practices and behaviours [9], enabling them to voluntarily make informed nutritional choices [10] to promote healthy living and improve nutritional status.

Rural areas face disadvantages in terms of access to information, knowledge and educational opportunities, both formal and informal [11], along with higher levels of poverty and food insecurity [12]. Targeted NE programmes effectively bridge the knowledge gap in rural settings [13], by improving dietary practices and overall well-being [14]. In addition, NE plays a crucial role in addressing extreme poverty and hunger, reducing child mortality, enhancing maternal health [15], empowering disadvantaged individuals with knowledge, social engagement and community engagement. Women, as key household decision-makers, serve important roles in childcare, food preparation, and resource management [16]. NE can equip women to make informed choices, utilise available resources wisely and adopt suitable nutrition practices, even in economically challenging environments. Achieving meaningful changes in dietary behaviour requires respect for cultural contexts and individual autonomy [17]. Evidence-based interventions and community-driven approaches offer valuable insights fostering supportive environments and improving the effectiveness of NE [18,19].

In this study, community-integrated NE was developed and implemented within a community-based framework to address prevailing nutritional issues. This follow-up intervention aimed to create opportunities for social support by involving relevant stakeholders, ensuring the success of the programs and promoting the application of acquired knowledge throughout the community. Despite socio-economic challenges such as poverty, limited education access and prevalent diseases, women play pivotal roles in families and communities worldwide [20,21]. Furthermore, women often carry a heavy burden of household responsibility, which can exacerbate their vulnerability [22] while significantly influencing and shaping food and nutritional behaviours [23]. Therefore, this study focuses on women.

## 2. Materials and Methods

This longitudinal study followed the Food and Agriculture Organisation framework for the development of a Nutrition Education Program (NEP) [23]. This study was organised into four phases.

### 2.1. Phase 1: Baseline Survey

An exploratory study was conducted to assess the prevalence of malnutrition and identify gaps in nutritional knowledge and behaviour among women (n = 75) [24].

### 2.2. Phase 2: Nutrition Education Program Formulation

A tailored NEP was developed, with effective tools designed to enhance nutrition knowledge and skills. The primary objective was to enhance overall nutrition knowledge and foster positive dietary behaviours.

### 2.3. Phase 3: NEP Implementation and Evaluation

The implementation phase design was a simple true intervention involving two equivalent groups tested for cause–effect relationships in a tightly controlled setting. The NEP was delivered to the intervention group (n = 35), while the control group (n = 30) did not receive the programme. The effectiveness of the implemented NE was evaluated by determining its short (immediate) impact on the nutritional knowledge of women in the intervention group.

### 2.4. Phase 4: Follow-Up

This was an intervention study design evaluating the long-term impact of the NEP on nutritional knowledge retention and behaviour (six months after intervention) in both the intervention (n = 32) and control groups (n = 30) [25]. This paper focuses on the impact of a nutrition education intervention and explores the changes in nutrition knowledge and behaviour over time (before and after the intervention). By isolating the intervention phase, we aimed to assess its effectiveness and provide insights into the role of NE in shaping behaviour.

This study aimed to evaluate the immediate and long-term impact of an NEP on the primary outcome of nutrition knowledge and the secondary outcome of nutrition behaviour. This study used an intervention design to assess the influence of an NEP on the retention of nutritional knowledge and nutrition-related behaviours. A quantitative research approach was utilised, involving the administration of questionnaires, which were applied during both the implementation and follow-up phases of this study to analyse and evaluate the outcomes.

This study followed the South African Medical Research Council (MRC) guidelines for human research [26]. Ethical approval was granted by the University of Witwatersrand Medical Ethics Committee for Human Research on 26 September 2008 (M080931). This study was part of the Integrated Nutrition Project in Qwa-Qwa (INPQQ) conducted by researchers from the Centre for Sustainable Livelihood (CSL) at Vaal University of Technology (VUT). All the study subjects participated voluntarily and each participant signed a written consent form after the project objectives and procedures had been explained and before the implementation of this study. Participant confidentiality was maintained throughout this study. No personal details were disclosed on any forms or during data capturing. Instead, unique identification numbers were assigned to each participant, corresponding to the measuring instruments, ensuring anonymity.

This study was designed as a follow-up to a previous research project that had implemented a nutrition education intervention for primary school children at a specific school, which was completed during 2010 [27]. In the current study, the focus shifted to female caregivers of children attending the same school. This approach aimed to enhance the nutritional knowledge and dietary behaviours of women to maximise the impact of the intervention. The goal was to reinforce earlier interventions by delivering similar nutrition messages (revised South African Food Based Dietary Guidelines (FBDG)) [28]. All female caregivers (n = 109) listed in the school register were invited to participate in this study. Of these, 96 caregivers responded to the invitation, but only 65 consented to participate in the baseline phase of this study. The study sample (n = 65) was then divided into two distinct groups, the intervention (treatment) group and the control group [29], resulting in a total study implementation phase sample size of n = 65, with the intervention group consisting of 35 participants and the control group including 30 participants. This division adhered to established scientific practice, which typically requires a minimum sample size of 30 subjects [30]. During the follow-up phase, three participants dropped out, leaving a total of 62 participants (control group: n = 30, intervention group: n = 32) who remained involved in this phase of this study. This study formulated a well-coordinated and systematic NEP with sufficient knowledge and time delivered through multiple channels [28]. A tailor-made group-based programme that underscored the importance of social relevance to reinforce a strong support network through group discussions and sharing of personal experiences was implemented. NE was implemented using two distinct approaches. The first approach involved imparting relevant, sufficient knowledge and skills to empower caregivers and motivating them to enhance their self-regulatory abilities and take appropriate actions. The NEP, as depicted in Table 1, was outlined as follows: (1) a theory-based class consisting of lecturers, group discussions and activities to enhance learning, facilitated in a preferred Sesotho language by competent facilitators; (2) experiential cooking classes, which included demonstrations, food tasting and food preparation skills, and these sessions were overseen by capable facilitators; and (3) the creation of vibrant booklets (pamphlets) and educational tools (such as food puzzles, flashcards and food models) to enhance the teaching and learning process. These visual aids were designed to engage the participants’ minds and maintain their attention [28]. Each lesson covered a specific nutrition topic, namely, (1) introduction to healthy eating and food guide, (2) guidelines for healthy eating, (3) healthy menu patterns, and (4) food safety and quality. Each nutrition topic lesson was carried out over four hours. Furthermore, cooking topics were discussed in conjunction with the nutrition topics and included (1) healthy affordable meals, (2) what is for lunch, (3) mixed meals and (4) healthy lunch box, respectively. The total time of presentation for each cooking topics was five and half hours. All existing cultural practice data that may influence dietary behaviour, food intake and nutritional knowledge of participants and their households were considered to ensure the development of culturally acceptable NE strategies. The study of NE development, including its theories and conceptual frameworks, is not described in this study.

The FAO nutrition education framework implementation methods were applied [23], including the creation of educational materials, caregiver training, execution of the training programme and evaluation. To assess the immediate impact of the implemented NE on the nutritional knowledge of caregivers in the intervention group, a Nutrition Knowledge Questionnaire (NKQ) was used. The intervention was facilitated by the researcher, who was knowledgeable about nutrition and a fluent South Sotho speaker. Additionally, four B-tech students were recruited and trained to provide support during theory and cooking classes, as well as during data collection. The intervention was conducted between April and May 2014, 6 months after the baseline survey was conducted. The inclusion and exclusion criteria were classified as female caregivers, no age restriction and having a child or children at the selected primary school residing in the study area. The exclusion criteria included male caregivers and caregivers with no primary school children in the area. A methodical strategy was used, involving the manipulation of one variable while maintaining control over the other [31]. NE intervention was exclusively administered to the intervention group and served as the sole intervention for this study, with no supplementary programmes introduced in conjunction with NE.

In both the implementation phase and the follow-up evaluation, two measuring tools were employed: (1) an NKQ was utilised to assess the participants’ nutritional knowledge during both the implementation and follow-up phases. The NKQ was specifically developed by the researcher (QEM Mangwane) for this study. (2) A dietary diversity questionnaire (DDQ) was employed during baseline survey and follow-up to gauge nutrition-related behaviour, which was a previously standardised and validated questionnaire [27,32].

Data were collected during both phases of this study. In May, immediately following the intervention, data collection was limited to the intervention group (n = 32). In the follow-up phase, data were collected in November, six months after the intervention. Data collection was performed at the same location and concurrently for both the intervention (n = 32) and control groups (n = 30). One participant (n = 1) in the control group did not complete the DDQ, resulting in a sample size of 29. Trained fieldworkers conducted one-on-one interviews in South Sotho and recorded the data using each measurement tool.

A variety of statistical tests were used, including Pearson’s chi test and paired *t*-test rank-sum (Mann–Whitney) test, to determine the degree and type of relationship between any two or more quantities (variables) and how they vary together over a period (seven days) [33]. The significant correlations between the variables in the two groups (intervention and control groups) and between different phases (baseline, post and follow-up phases) were tested. Statistical significance was set at 0.05 (*p* ˂ 0.05). Data analysis was performed using the Stata statistical software (StataCorp. 2009, Version 11, College Station, TX, USA: StataCorp LP) following the collection and cleaning of data on a Microsoft Excel spreadsheet. The analysis used descriptive statistics, including means and standard deviations (SD), frequencies (f) and percentages (%).

The overall nutrition score was calculated to assess nutritional knowledge. The questions from the NKQ were categorised into six relevant nutrition topics for this study: FBDG, unhealthy food, daily meal frequency, portion size, food composition and hygiene. The overall nutrition knowledge score was determined by adding the number of correctly answered questions from each nutrition topic and then dividing the total by six (number of topics, n = 6). The Dietary Diversity Score (DDS) was computed by summing the number of food groups consumed over seven days [34]. In this calculation, the food items were not quantified. Each participant received one point for each of the nine food groups consumed over the seven days, with a maximum of nine points awarded if all food from all nine groups was consumed [35]. Dietary diversity tertiles were established based on the nine food groups and 101 food items (Food Variety Score (FVS)) to differentiate between low-, medium- and high-diversity diets. The tertile ranges for dietary diversity were categorised as low (0–3), medium (4–5) and high (6–9), whereas food variety was assessed as low (<30), medium (30–60) and high (>60). These categories are consistent with those reported in previous studies conducted in South Africa [32].

## 3. Results

In Table 2, the nutrition knowledge overall score shows the percentage of correct answers across the six nutrition topics, both before and after the intervention. The intervention group displayed higher overall knowledge scores (63.7%), indicating an increase in knowledge across all six nutrition topics immediately after the intervention. In the category ‘FBDG knowledge’, the results demonstrate statistically significant improvements in FBDG knowledge for the intervention group. Notably, there were significant differences in messages related to fruit and vegetable consumption (*p* = 0.02) as well as significant differences between the groups for messages about hard fat use (*p* < 0.001), sugary foods (*p* = 0.005) and salty/fatty foods (*p* = 0.005). The category of ‘unhealthy food item knowledge’ revealed statistically significantly higher levels of knowledge in the intervention group after the intervention. This is particularly evident in messages about the use of creamers (*p* = 0.005), hard fat (*p* < 0.001) and the importance of avoiding the addition of salt and fat to starchy foods (*p* = 0.005). Similarly, for ‘meal frequency knowledge’, a significant difference (*p* = 0.001) in knowledge between the study groups after the intervention was noted. The intervention group showed better improvement in knowledge regarding meal frequency following the intervention. For ‘food portion size knowledge’, the findings reveal significant disparities in knowledge levels between the two study groups, particularly in relation to portion sizes of eggs (*p* = 0.01), cooked vegetables (*p* = 0.02), rice (*p* = 0.05) and maas (*p* = 0.02). Knowledge regarding all portion sizes (n = 7) increased in the intervention group after the intervention; however, this was not statistically significant. The findings for ‘food composition knowledge’ showed improvement in all categories (n = 4), although the increase did not reach statistical significance. For the category ‘hygiene knowledge’, the knowledge levels increased within the intervention group in relation to both personal and food hygiene following the intervention; however, the increase was not statistically significant, which may have been attributed to the relatively high baseline knowledge level in this area. The participants demonstrated good initial hygiene knowledge, leaving limited room for measurable improvement. The lack of statistical significance may also have been contributed by the intensive focus on topics such as dietary diversity and food preparation, leading to less observable changes in knowledge.

In the follow-up results, as depicted in Table 3, the intervention group exhibited significantly higher overall knowledge scores (64.4%) and a mean knowledge improvement of 15.3 percentage points, indicating sustained knowledge improvement six months after the intervention. The results show improved knowledge across all FBDG topics in the experimental group at follow-up. In the intervention group, there was significant knowledge improvement for drinking water and vegetable consumption messages between baseline and follow-up. Furthermore, the results for hygiene knowledge indicate a post-intervention increase in knowledge. Notably, the intervention group exhibited significantly greater knowledge levels in the personal hygiene and food safety domains during the follow-up assessment.

The dietary diversity, as seen in Table 4, shows a modest increase in dietary variety within the intervention group. The post-intervention means were slightly higher (DDS, 8.75; FVS, 37.6) compared to the pre-intervention values (DDS, 8.51; FVS, 36.8). Notably, there was a significant improvement in the FVS after the intervention (*p* = 0.012).

There was a significant difference in food group diversity (*p* = 0.029) and food variety (*p* = 0.001) between the study groups after the intervention (Table 5). The intervention group consumed a slightly more diverse diet, including 6 to 9 food groups and 30 to 60 food items.

Table 6 shows a mixed pattern in food group variety. In the intervention group, the variety of some food groups improved, while the variety of others decreased between the pre- and post-intervention phases.

Table 7 presents the significant change (*p* = 0.020) in the number of food items consumed between the pre- and post-intervention phases. Additionally, there was a significant difference (*p* = 0.001) in the number of food items consumed between the two study groups following the intervention. The majority (71.9%) of the intervention group consumed between 31 and 50 food items, suggesting healthy food consumption behaviour post-intervention.

## 4. Discussion

The study results indicate a significant increase in nutrition knowledge, sustained retention of that knowledge over the long term, and a transition towards healthier dietary choices, all suggesting the positive influence of NE.

### 4.1. Nutrition Knowledge

The findings of this study indicated improvements in knowledge following the intervention. The comparison of means for pre-intervention (49.1%), post-intervention (63.7%) and follow-up (64.4%) reflected a knowledge improvement of 15.3% between pre- and post-intervention. This suggests a positive influence of NEP studies on knowledge levels. Similar knowledge increases after the intervention were reported in local [36] and international [37] studies. The current study findings indicate that an integrated NEP that aimed to address nutrition knowledge and behaviour was successful in elevating nutrition knowledge in disadvantaged communities with limited resources. These findings confirm the widely documented success of NE in increasing the level of nutrition knowledge and other nutrition-related behaviours [38,39]. Follow-up overall knowledge mean improvement (63.7–64.4%) and higher (62.2%, 65.2%, 86.7%, 59.2%, 52.9% and 60.0%) overall scores in all nutrition topics (n = 6) indicate positive long-term knowledge retention in the intervention group. These findings concur with those of a previous study that reports higher retention scores at follow-up [40]. However, the results contradict the findings of another study that reports poor knowledge retention in the intervention group at follow-up [41]. The present study clearly illustrates sustained knowledge retention within the intervention group during the follow-up period, indicating that female caregivers possessed a solid foundation of nutrition knowledge six months after the intervention. This enduring knowledge retention likely results from the comprehensive educational approach employed, which has previously been shown to enhance knowledge retention [40]. Therefore, the long-term knowledge retention observed in the current study can be attributed to the comprehensive learning methods implemented during NE intervention. Knowledge in the intervention group improved in all nutrition topics (n = 6); however, the improvement was only significant for FBDG messages related to the daily consumption of fruits and vegetables. Another significant FBDG knowledge improvement was observed for messages related to drinking water and fruit consumption at follow-up. A similar significant increase in knowledge related to essential food group consumption messages over the long term has been reported [42]. Another study involving interventions for both parents and children documented substantial improvements in maternal knowledge and practices following the implementation of NE [43]. These outcomes underscore the efficacy and sustainability of NE in enhancing nutritional awareness aligned with national dietary guidelines, particularly in resource-constrained settings. The increase in knowledge mean (49.1–63.7) immediately following the intervention indicated a favourable short-term effect of NE on nutrition knowledge, mirroring the effectiveness of this intervention. These findings align with those of other studies that documented positive outcomes following similar interventions [44]. The literature reflects how an immediate positive change in knowledge and improvement in knowledge was associated with the beneficial impact of regular interventions aimed at refreshing knowledge and practice in nutrition [39]. In the present study, NE was meticulously structured with four distinct theory classes, each focusing on a specific nutrition topic, held over four weeks. The noticeable enhancement in nutrition knowledge immediately following the intervention can thus be attributed to the effective implementation of a structured educational nutrition intervention.

This study’s good knowledge scores in various nutrition topics, including FBDG, suggest the potential for adherence to dietary guidelines and positive shifts towards healthier behaviours. In turn, this may contribute to addressing the root causes of malnutrition in the study population. Specifically, the FBDG knowledge score at the follow-up assessment (62.2%) surpassed the national baseline knowledge percentages reported in the South African National Health and Nutrition Examination Survey (SANHANES), which were 54.3% in urban formal, 50.5% in urban informal, and 49.7% in rural informal settings [45]. The current study results demonstrate that 62.2% of female caregivers maintained a strong grasp of dietary guidelines in the long term. Strong maternal knowledge was associated with a higher likelihood of adhering to nutritional guidelines and being motivated to cultivate and sustain healthy dietary habits [46,47,48]. The current study’s knowledge improvement and retention concur with a long-term effect on knowledge that was reported in a controlled intervention study [41] despite contradictions from the reported deficits of knowledge retention in the long-term in other studies [49]. The study findings confirm the important role of an NEP in elevating nutrition knowledge in disadvantaged communities with limited resources. A comparison between the intervention group and the control group revealed higher overall knowledge scores in the intervention group (63.7%) compared to the control group (36.3%) after the intervention. Knowledge level differences between the study groups were significant for the topics of egg (*p* = 0.01), cooked vegetables (*p* = 0.02), rice (*p* = 0.05) and portion size (*p* = 0.02) knowledge; and regarding unhealthy food for messages related to the use of creamer (*p* = 0.05), hard fat (*p* = 0.005), sugary foods (*p* = 0.001) and salty foods (*p* = 0.005) and meal frequency messages (*p* = 0.001). Similar disparities in knowledge, with higher knowledge levels observed in the intervention group post-intervention compared to the control groups, have been documented in other studies [50,51]. The findings in the present study’s control group corroborate the presence of limited nutrition knowledge in socioeconomically disadvantaged communities [52]. The findings also highlight the vital role of NE in enhancing nutrition knowledge within disadvantaged communities. Maternal nutrition knowledge significantly influences nutritional practices among socioeconomically disadvantaged women [53,54]. The low knowledge levels observed in the control group may adversely affect nutritional behaviour, leading to poor health and nutrition outcomes in low-income households. This is particularly concerning given the links between dietary behaviours and the risks of malnutrition and various non-communicable diseases [54].

Adequate knowledge, comprehension of appropriate practices, development of skills and establishment of good behaviours are fundamental for managing nutrition-related issues effectively [40]. The current study’s findings reveal an improvement in overall unhealthy knowledge scores (49.1% to 65.2%) and the retention of knowledge related to unhealthy food messages within the intervention group at the follow-up stage. The observed improvement in awareness of unhealthy foods within this study may play a crucial role in influencing the consumption of food items that should be limited, potentially enhancing nutrition and health outcomes among the study population. However, it is worth noting that some other studies report no significant improvement in dietary behaviour, even when individuals have adequate knowledge [55]. This study’s intervention results indicated a positive knowledge change in the intervention group in relation to both hygiene questions. Personal and food hygiene knowledge improved from 37.0 to 53.8% and from 45.9 to 51.7%, respectively, in the intervention group between pre- and post-intervention. The results from this study are similar to others [56], whereby high post-intervention food safety knowledge scores occurred in the intervention group. Positive hygiene practices were reported among individuals with high hygiene knowledge in communities with inadequate water and sanitation facilities [57]. Therefore, the evident hygiene knowledge improvement in the present study may translate to an improved attitude and behaviour towards hygiene, mitigating the poor hygiene circumstances in this disadvantaged community.

Mixed results were observed in the intervention group after the intervention. This study’s findings showed that the FBDG presented a varied picture, with some questions showing improvement (n = 6), while others exhibited a decline (n = 7) in the intervention groups at follow-up. This finding concurs with the poor knowledge retention reported in intervention studies [37,39]. Knowledge levels improved for nearly all 10 FBDG messages after the intervention; however, a negative change in knowledge was observed in the same group. Specifically, knowledge levels for messages related to drinking water decreased from 45.6 pre-intervention to 44.1 post-intervention in the experimental group, although this change did not reach statistical significance.

### 4.2. Nutrition Behaviour

The present study’s diversity mean improvement post-intervention (DDS 8.75, FVS 37.6) reflects a positive change in dietary behaviour in this under-resourced community.

### 4.3. Dietary Habits

The findings related to diet diversity in this study indicate a positive shift in dietary behaviour following the intervention, particularly within the intervention group. A comparison of the pre- and post-DDS values (7.77 and 8.75) and FVS values (36.8 ± 11.7 and 37.6 ± 9.72) revealed a more diverse diet at the follow-up stage (as shown in Table 3 and Table 5). Diversity improved from DDS 7.76 and FVS 36.8 before the intervention to DDS of 8.75 and FVS of 37.6 at follow-up, suggesting a growing trend towards the consumption of a wider variety of food groups post-intervention.

Similar improvements in diet quality have been reported in both local [57] and international [48] studies after similar interventions. Consuming a diverse diet, encompassing various food groups, is known to enhance nutrient intake and dietary quality, promote nutritional status and offer protection against nutrition-related issues [58]. Therefore, the observed improvement in diet diversity is likely to have a positive impact on diet quality, health and nutritional status in this study’s context. The findings revealed significant differences in dietary habits among the study groups. The intervention group exhibited notably higher food group diversity than the control group, with a *p*-value indicating no significant difference (*p* = 0.29). In the intervention group, all participants (n = 32) were classified as having high diversity, consuming between seven and nine food groups. Furthermore, the classification of the FV category showed a significant difference (*p* = 0.000) between the two groups, with the intervention group showing better food variety (medium variety) than the control group, which had a poor variety (low variety). These findings demonstrate that a tailor-made NEP intervention that aims to improve knowledge and related nutritional behaviour can effectively enhance household dietary diversity [59], especially in poor communities where monotonous diets are consumed and malnutrition is high [60].

The diet diversity results from this study revealed a mixed pattern during follow-up. The intervention group variety improved significantly in the meat (*p* = 0.004), cereal (*p* = 0.000), egg (*p* = 0.021), dairy (*p* = 0.020) and oil and fat (*p* = 0.020) groups. However, there was a significant decline in variety observed for ‘other’ fruits group (*p* = 0.001) and the ‘other’ vegetable group (*p* = 0.005) at follow-up. The findings discussed above highlight the shortcomings of NE interventions in maintaining dietary behaviour, particularly within under-resourced households. Despite adequate nutrition knowledge, the results indicate challenges in sustaining positive dietary changes over time. Similar findings were reported in a local intervention study [61] that adapted a diabetes nutrition education program for adults with type 2 diabetes from a primary to a tertiary healthcare setting.

The summary of FVS (±SD) for all foods consumed from all nine food groups (N = 9) indicated moderate food variety within the food group after intervention in the intervention group. Pre- and post-intervention food group diversity total means of 33.1 and 37.6 indicate a 4.5 mean improvement in food group diversity post-intervention with a variety increase in the meat (7.15 ± 2.35), dairy (3.76 ± 1.19), egg (1 ± 0), cereal (8.78 ± 2.09), legume (2.86 ± 0.94) and fats and oil (2.12 ± 0.55) food groups (Table 6). Dietary diversity is defined as the number of individual food items or food groups consumed over a period [34,62]. This study findings indicate better variety and diet diversity improvement in the intervention group post-intervention, reflecting a positive effect of intervention on the diet quality in low-income households. Many intervention studies report similar positive changes in dietary habits and a variety of improvements post-intervention [13,63]. The introduction to a variety of nutritious foods and limiting non-core food exposure from an early age are important strategies for improving later diets [64]. The inclusion of a variety of foods by female caregivers in this study may influence children’s eating behaviour towards healthy food preferences and healthy food choices, thus improving diet quality and nutrition outcomes.

### 4.4. Food Consumption Behaviours

A follow-up assessment revealed improved food consumption patterns following the intervention, particularly in the consumption of six food groups. Post-intervention, the consumption of the meat group (7.15 ± 2.35), eggs (1 ± 0), dairy (3.76 ± 1.19), cereal (8.78 ± 2.09), legumes (2.86 ± 0.95) and fats and oils (2.12 ± 0.55) increased significantly. It is worth noting that these improvements occurred in food groups that were previously rarely consumed, underscoring the effectiveness of the NEP in reshaping the food choices and eating behaviours of women from disadvantaged households.

In South Africa, the limited inclusion of dairy products in the daily diet is a well-known issue, especially in disadvantaged households [65]. Additionally, the consumption of legume food groups was lacking in the study population [27]. However, following the intervention in the current study, significant changes in food intake patterns were observed with noticeable improvements in the consumption of legumes and dairy food items. Similar increases in the intake of legumes, dairy and vegetables were reported after a six-month nutrition intervention in São Paulo [48]. The findings from the intervention group’s food group diversity suggest a positive impact of the intervention on the consumption behaviour of women in a low-income community, particularly in terms of improving the intake of meat, dairy, legumes, and egg groups. The observed high means of food consumption from the cereal (7.7 ± 2.8–8.78 ± 2.09), meat (6.4 ± −2.8 7.15 ± 2.35) and other vegetables (6.3 ± 2.8–5.40 ± 1.82) food groups pre- and post-intervention suggest that the participants in this rural study maintained a typical diet that emphasises starchy food, meat and vegetables. This finding concurs with those of previous studies that reported a diet dominated by starchy food with the inclusion of cheap meat sources and vegetables in under-resourced SA households [66].

The results related to food consumption highlighted a significant change (*p* = 0.020) in the number of food items consumed before and after the intervention. Before the intervention, a considerably higher number of food items was consumed, ranging from 11 to 70 food items, as opposed to 11 to 60, post-intervention. Furthermore, there was a noticeable difference in the number of food items consumed between the two study groups after the intervention, with a significantly higher number of food items consumed in the intervention group than in the control group (*p* = 0.001). All nine food groups were incorporated into the daily diet of participants in the rural intervention group. This aligns well with the South African Food Based Dietary Guidelines (SAFBDG), which recommends the message ‘eat a variety of food’ to promote healthy eating and nutrition. The consumption patterns observed in the current study reflect healthy food intake that is consistent with these dietary guidelines.

However, it is worth noting that there is evidence of frequent consumption of certain food items that may be detrimental when consumed excessively, such as custard, powdered milk, drinking yoghurt and processed meat. These food items contain high levels of fat, sugar and salt, which can contribute to various nutrition-related diseases [67]. The South African Food Based Dietary guidelines recommend restricted consumption of foods high in fat, salt and sugar for optimal health [6]. This NEP study targeted existing knowledge gaps and unhealthy nutritional behaviours among female caregivers using FBDG messages. The intervention successfully increased and maintained nutrition knowledge in the current study group from 58.3% to 62.2%.

Some constraints may have impacted this study’s outcomes, and include the small sample size of caregivers, which cannot allow for the results to be generalised to represent all women caregivers in SA. Furthermore, the socio-economic constraints may have impacted the practical application of improved eating practices, even though significant changes occurred with the nutrition knowledge. Lastly, the self-reported data from the participants were reliant on their ability to recall their food intake patterns.

## 5. Conclusions

The NEP improved nutrition knowledge and shifted food consumption patterns towards a positive healthy nutritious diet. These findings emphasise the value of a well-designed, socially appropriate NEP that uses cross-curricula and experimental learning approaches as a strategy to facilitate healthy nutritional behaviour among women in poorly resourced communities. Evidence of persistent unhealthy nutrition behaviour in the presence of high nutrition knowledge concludes that nutrition knowledge alone is not sufficient to change nutrition behaviour in disadvantaged communities.

Knowledge improvement and long-term knowledge retention among women in this study concluded that nutrition intervention can successfully improve knowledge to compensate for the lack of knowledge in women living in under-resourced communities. On the other hand, evidence of knowledge gaps as well as gaps between knowledge and behaviour in this study post-intervention confirms the shortcomings of nutrition intervention in improving knowledge and nutrition-related behaviour in these communities. Nevertheless, the research results offer a significant perspective, indicating the necessity for a comprehensive nutritional approach that tackles socioeconomic obstacles in under-resourced communities.

Addressing malnutrition requires a combination of strategies which go beyond just the dissemination of information, but integrate multifaceted approaches. Education alone will not drive sustainable dietary improvements. Combined approaches such as community gardening, funding support and policy-driven food security programs can amplify the benefits of NEPs.

## Figures and Tables

**Table 1 nutrients-16-04107-t001:** NEP design summary.

Week	Time	Lecture Topic	Activities	Time	Cooking Topics	Activities	Total Time
1	90 min	Introduction to healthy eating and food guide	Lecture Food display Discussions Evaluation	4 h	Healthy affordable meals	Discussions Demonstration Cooking Food tasting	5 h 30 min
2	90 min	Guidelines for healthy eating	Lecture Food display Food tasting Evaluation	4 h	What is for lunch	Discussion Demonstration Cooking Food tasting	5 h 30 min
3	90 min	Healthy menu patterns	Lecture Planning menus Discussions Evaluation	4 h	Mixed meals	Discussion Demonstration Cooking Food tasting	5 h 30 min
4	90 min	Food safety and quality	Lecture Demonstration Display Evaluation	4 h	Healthy lunch box	Discussion Demonstration Cooking food tasting	5 h 30 min

**Table 2 nutrients-16-04107-t002:** Study groups’ overall score of correct knowledge pre- and post-intervention.

PARAMETERS	Correct Answers Pre (%)	Correct Answers Post (%)	Correct Answers Pre (%)	Correct Answers Post (%)
	Experimental Group		Control Group	
**1. FBDG knowledge**				
Do you think children should eat different kinds of food?	45.7	53.8	54.3	46.2
Starchy food like rice, should be included in every meal	44.6	54.4	55.4	45.6
Legumes like dry beans should be eaten regularly	50.0	57.1	50.0 ^a^	42.9 ^a^
Milk, maas or yoghurt should be eaten daily	45.0	49.1	55.0 ^a^	50.9 ^a^
Meat, such as fish, chicken, and beef should be eaten daily	30.0	52.9	70.0	47.1
Fruits and vegetables should be eaten daily	46.1 ^a^	54.6 ^a^	53.9	45.4
Drink 8 glasses of water daily	45.6	44.1	54.4	55.9
Sweets and cold drinks should be eaten between meals when hungry	50.0	70.6 ^b^	50.0	29.4 ^b^
An active person is a healthy person	47.9	53.1	52.1	46.9
**2. Unhealthy food items knowledge**				
Milk rather than creamer should be used in coffee and tea	53.9	65.0 ^a^	46.1	35.0 ^a^
Using hard fat is better than using vegetable oil	44.8	75.8 ^a^	55.2	24.2 ^a^
The intake of fruit juice and sports drinks should be limited	47.7	53.5	52.3	46.5
Fat or salt should be added to starchy foods to make them more nutritious	50.0	76.0 ^a^	50.0	24.0 ^a^
**3. Eating patterns correct answers**				
How many meals should a child eat per day?	66.7	87.1 ^b^	33.3 ^a^	12.9 ^ab^
**4. Portion size knowledge**				
2 slices of bread	47.1	68.2	52.9	31.8
1 egg	48.9	59.3 ^a^	51.1	40.7 ^a^
1 cup cooked vegetables	52.6	92.3 ^a^	47.4	7.7 ^a^
A piece of meat the size of your palm	42.9	54.7	57.1	45.3
1/2 cup of rice	40.5	61.8 ^a^	59.5	38.2 ^a^
1 cup of maas (fermented milk)	46.2	50.8 ^a^	53.8	49.2 ^a^
1 medium fruit	46.7	52.5	53.3	47.5
**5. Food composition knowledge**				
Starchy foods are the main source of energy	42.6 ^a^	53.1	57.4 ^a^	46.9
Vegetables and fruit are good sources of vitamins, minerals and water	45.8	53.9	54.2	46.1
Fish, chicken, meat and eggs are protein-rich foods	47.8	52.4	52.2	47.6
Meat contains lots of saturated fat	45.8	54.9	54.2	45.1
**6. Hygiene knowledge**				
Wash hands before cooking if you used the toilet	37.0	53.8	63.0	46.2
Wash fruits and vegetables before eating	45.9	51.7	54.1	48.3

^a^ Same superscript letters indicate between-group significant differences, statistically significant if *p* < 0.05; ^b^ Same superscript letters indicate between-group significant differences, statistically significant if *p* < 0.05; Abbreviation: FBDG, food-based dietary guidelines.

**Table 3 nutrients-16-04107-t003:** Overall score of correct knowledge at pre-intervention and follow-up.

Parameter	Intervention n = 35	Intervention n = 32	Control n = 30
Nutrition Knowledge Topics	Pre-Intervention (%)	Follow-Up (%)	Follow-Up (%)
FBDG knowledge	45.4	62.2	37.8
Knowledge of unhealthy food	49.1	65.2	34.8
Eating pattern knowledge	66.7	86.7	13.3
Food portion size knowledge	46.4	59.2	40.8
Food composition knowledge	45.5	52.9	47.1
Hygiene knowledge	41.5	60.0	40.0
Overall mean	49.1	64.4	35.6

**Table 4 nutrients-16-04107-t004:** Intervention group dietary diversity pre- and post-intervention.

Outcomes Measure	Study Phase	Sample Size	Mean	Range	*p*-Value
DDS	Pre-intervention	n = 35	8.51	7–9	0.075
	Post-intervention (6 months)	n = 32	8.75	7–9	
FVS	Pre-intervention	n = 35	36.8	21–68	0.012 *
	Post-intervention (6 months)	n = 32	37.6	19–49	

* indicates significance; Abbreviation: DDS, dietary diversity scores; FVS, food variety scores.

**Table 5 nutrients-16-04107-t005:** Study groups’ dietary diversity score categories after intervention.

Outcomes Measure	Study Groups	DDS Tertiles			*p*-Value
Food groups (n = 9)	DDS	Low diversity (1–3)	Medium diversity (4–5)	High diversity (6–9)	
Post- intervention (n = 61)	Intervention group (n = 32)	n = 0 (0%)	n = 0 (0%)	n = 32 (100%)	0.029 *
	Control group (n = 29)	n = 0 (0%)	n = 1 (3.5%)	n = 28 (96.5%)	
Food items (n = 101)	FVS	Low variety 0–29	Medium Variety 30–60	High Variety 61–101	
Post- intervention (n = 61)	Intervention group (n = 32)	n = 6 (18.8%)	n = 26 (81.2%)	n = 0 (0%)	0.000 *
	Control group (n = 29)	n = 22 (75.8%)	n = 7 (24.1%)	n = 0 (0%)	

* indicate significance; Abbreviation: DDS, dietary diversity scores; FVS, food variety scores. Comparison of dietary diversity categories between groups post-intervention, statistically significant if *p* < 0.05. Data are presented in three tertiles (low, medium, high).

**Table 6 nutrients-16-04107-t006:** Intervention group food group variety pre-and post-intervention.

Experimental Group	Pre-Intervention (n = 35)		Post-Intervention (n = 32)		*p*-Value
Outcomes Measure	Mean ± SD	Range	Mean ± SD	Range	
Meat group	6.4 ± 2.8	3–12	7.15 ± 2.35	3–12	0.004 *
Egg group	0.7 ± 0.0	0–1	1 ± 0	1–1	0.021 *
Dairy products	3.4 ± 2.0	0–8	3.76 ±1.19	2–6	0.020 *
Cereal, roots and tubers group	7.7 ± 2.8	4–14	8.78 ± 2.09	4–13	0.00 *
Legumes and nuts	2.0 ± 0.0	0–5	2.86 ± 0.95	1–5	0.430
Vitamin A-rich fruits and vegetables	4.6 ± 1.8	0–8	3.21 ± 1.36	1–6	0.130
Other fruits	7.0 ± 2.2	0–10	3.96 ± 2.00	0–10	0.001 *
Other vegetables	6.3 ± 2.8	1–12	5.40 ± 1.82	2–10	0.005 *
Oil and fats	1.9 ± 0.8	0–3	2.12 ± 0.55	1–4	0.020 *
**TOTAL**	**36.8 ± 11.7**	**11–68**	**37.6 ± 9.72**	**32–59**	**0.012**

* Indicate significance; Abbreviation: SD, standard deviation. Data are presented as mean ± SD. Mean differences show pre- and post-measure phases. Comparing food group variety within the intervention group at different study phases, statistically significant if *p* < 0.05.

**Table 7 nutrients-16-04107-t007:** Summary of consumed food items before and after intervention.

	Pre-Intervention	Post-Intervention		
Number of Food Items Consumed (N = 101)	Pre-Intervention (n = 75)	Intervention Group (n = 32)	Control Group (n = 29)	*p*-Value
0–10	n = 0 (0%)	n = 0 (0%)	n = 1 (3.5%)	0.020 *
11–20	n = 6 (8.0%)	n = 1 (3.1%)	n = 13 (44.8%)	
21–30	n = 21 (28.0%)	n = 6 (18.7%)	n = 8 (27.6%)	
31–40	n = 20 (26.7%)	n = 12 (37.5%)	n = 4 (13.8%)	
41–50	n = 17 (22.7%)	n = 11 (34.4%)	n = 3 (10.3%)	
51–60	n = 7 (9.3%)	n = 2 (6.3%)	n = 0 (0%)	
61–70	n = 4 (5.3%)	n = 0 (0%)	n = 0 (0%)	
** *p* ** **-value**		0.001 *		

* indicates significance; Comparing the total amount of consumed food items within and between groups, statistically significant if *p* < 0.05.

## Data Availability

The data supporting the findings of this study are available upon reasonable request from the corresponding author mangwaneqem@tut.ac.za. Access to the data will be provided in accordance with ethical guidelines and institutional policies.

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
