# Peer review of "Impact of a Nutrition Knowledge Intervention on Knowledge and Food Behaviour of Women Within a Rural Community"

_nutrients, 2024, doi:10.3390/nu16234107_

Round 1
Reviewer 1 Report
Comments and Suggestions for Authors
Thank you for the opportunity to review the very interesting and informative work entitled "Impact of a nutrition knowledge intervention on knowledge and food behavior of women within a rural community" by Queen Elizabeth Mangwane, Abdulkadir Egal and Delia Oosthuizen
The topic taken up by the authors in my opinion is very important for public health. Proper, personalized health education, as the authors have shown, brings satisfactory results. The results of these studies, although several years have passed since their acquisition, can still be an important argument in taking action to improve nutritional knowledge not only in the rural community, but also in other regions.
While reading the paper, I noticed minor errors that required checking/correction:
1). page 4, paragraph 2, sentence "In both the implementation phase and the follow-up evaluation, two measuring tools were employed: (1) an NKQ was used to assess participants' nutritional knowledge during both the implementation and follow-up phases. The NKQ was specifically developed by the researcher for this study" . Should it be understood that one person among the authors participated in developing the questionnaire?
2). The Table 1 - section "unhealthy food items knowledge" in the sentence "Fat or salt...." the repeated phrase "should be added" should be removed
3). Please explain or correct the word "maas" in table 1 in the phrase "Milk or maas of yoghurt...";
4). in the chapter “Discussion Nutrition knowledge”, line 3 - after the number 15.3 there should be a % sign.
5). Chapter Conclusion. 3rd paragraph "The purpose of this study..." is not necessary.
Author Response
Comment 1: page 4, paragraph 2, sentence "In both the implementation phase and the follow-up evaluation, two measuring tools were employed: (1) an NKQ was used to assess participants' nutritional knowledge during both the implementation and follow-up phases. The NKQ was specifically developed by the researcher for this study" . Should it be understood that one person among the authors participated in developing the questionnaire?
Response 1: responded and highlighted in yellow
Comment 2: The Table 1 - section "unhealthy food items knowledge" in the sentence "Fat or salt...." the repeated phrase "should be added" should be removed
Response 2: completed as requested
Comment 3: Please explain or correct the word "maas" in table 1 in the phrase "Milk or maas of yoghurt...";
Response 3: explanation included
Comment 4: in the chapter “Discussion Nutrition knowledge”, line 3 - after the number 15.3 there should be a % sign.
Response 4: symbol included
Comment 5: Chapter Conclusion. 3rd paragraph "The purpose of this study..." is not necessary.
Response 5: words removed
In addition, the spelling was checked against British English. Other errors corrected and highlighted, i.e. significance

Reviewer 2 Report
Comments and Suggestions for Authors
Impact of a Nutrition Knowledge Intervention on Knowledge and Food Behavior of Women within a Rural Community
Strengths:
Clear Purpose and Design: The study effectively addresses a critical issue—malnutrition in under-resourced communities—using a structured, quasi-experimental approach.
Relevant Outcomes: The research highlights significant findings, such as improvements in nutritional knowledge and dietary diversity, which align with the goals of the Nutrition Education Programme (NEP).
Comprehensive Data Presentation: The manuscript provides detailed data, including pre- and post-intervention comparisons, dietary diversity, and knowledge retention metrics.
Culturally Tailored Intervention: The program considers cultural contexts, utilizing Sesotho language facilitators and context-specific educational materials.
Weaknesses and Recommendations:
Abstract:
Weakness: The abstract is dense and lacks clarity in separating the methodology from the results and conclusions.
Recommendation: Rewrite the abstract to explicitly outline the study’s objectives, methods, key findings, and implications in a concise manner.
Introduction:
Weakness: Although the introduction establishes the study's context, it is overly descriptive and lacks critical synthesis of the literature.
Recommendation: Focus on framing the knowledge gap and the rationale for the study more succinctly. Limit historical background details to improve readability.
Methodology:
Weakness: The manuscript lacks a robust explanation of the intervention framework, including detailed curriculum content and delivery methods.
Recommendation: Provide a supplementary table summarizing the NEP’s curriculum, including duration, frequency, and specific activities conducted.
Weakness: Ethical considerations are mentioned but not elaborated upon.
Recommendation: Expand on how ethical principles were maintained, especially regarding informed consent and participant privacy.
Results:
Weakness: While results are presented comprehensively, the tables are crowded and may confuse readers.
Recommendation: Simplify or restructure tables by highlighting only statistically significant findings and grouping similar metrics.
Weakness: Limited focus on explaining non-significant findings.
Recommendation: Discuss why certain improvements, such as hygiene knowledge, did not achieve statistical significance.
Discussion:
Weakness: The discussion reiterates results without adequately exploring broader implications or limitations.
Recommendation: Critically examine the challenges of long-term behavior change and provide actionable recommendations for scaling the intervention.
Weakness: The manuscript does not address potential biases or limitations sufficiently.
Recommendation: Add a paragraph discussing limitations, including sample size constraints, generalizability, and self-reported data reliability.
Conclusion:
Weakness: The conclusion is overly optimistic and does not reflect the mixed results regarding sustained behavior change.
Recommendation: Revise to include balanced insights, emphasizing the need for complementary strategies beyond education.
References:
Weakness: There is inconsistency in citation formatting and some references appear outdated.
Recommendation: Update citations to include recent, relevant studies and ensure compliance with Nutrients formatting guidelines.
Language and Style:
Weakness: Certain sections, especially in the introduction and discussion, use repetitive and verbose phrasing.
Recommendation: Edit for conciseness and eliminate redundant statements to improve readability.
Overall Recommendation:
The manuscript addresses an important topic with practical implications and provides robust data to support its findings. However, improvements in clarity, methodological details, and discussion depth are necessary to enhance the paper's quality and impact. Once revised, the paper has the potential to make a valuable contribution to the literature on nutrition education in disadvantaged settings.
Author Response
Comment 1: Rewrite the abstract to explicitly outline the study’s objectives, methods, key findings, and implications in a concise manner.
Response 1: The authors reviewed and made minor corrections.
Comment 2: Focus on framing the knowledge gap and the rationale for the study more succinctly. Limit historical background details to improve readability.
Response 2: Corrections made and highlighted in yellow
Comment 3: Provide a supplementary table summarizing the NEP’s curriculum, including duration, frequency, and specific activities conducted.
Response 3: Table incorporated
Comment 4: Expand on how ethical principles were maintained, especially regarding informed consent and participant privacy.
Response 4: Included and highlighted in yellow.
Comment 5: Simplify or restructure tables by highlighting only statistically significant findings and grouping similar metrics.
Response 5: The authors respectfully disagree with your suggestion and prefer to leave the tables as is
Comment 6: Discuss why certain improvements, such as hygiene knowledge, did not achieve statistical significance.
Response 6: Two sentences have been incorporated.
Comment 7: Critically examine the challenges of long-term behavior change and provide actionable recommendations for scaling the intervention.
Response 7: Recommendations have been refined in the conclusion
Comment 8: Add a paragraph discussing limitations, including sample size constraints, generalizability, and self-reported data reliability.
Response 8: Included - last paragraph of section 4
Comment 9: Revise to include balanced insights, emphasizing the need for complementary strategies beyond education.
Response 9: As per response 7
Comment 10: Update citations to include recent, relevant studies and ensure compliance with Nutrients formatting guidelines.
Response 10: reference list corrected as per journal guidelines
Comment 11: Edit for conciseness and eliminate redundant statements to improve readability.
Response 11: Manuscript was submitted through Grammarly, language edited and checked again for grammar and spelling errors through WORD, using British English
